# Effectiveness of mRNA COVID-19 vaccine booster doses against Omicron severe outcomes

Ramandip Grewal [1], Lena Nguyen[2], Sarah A. Buchan[1,2,3,4], Sarah E. Wilson[1,2,3,4], Sharifa Nasreen [2,3], Peter C. Austin[2,5], Kevin A. Brown[1,2,3], Deshayne B. Fell [2,6,7], Jonathan B. Gubbay[1,8], Kevin L. Schwartz [1,2,3], Mina Tadrous[2,9,10], Kumanan Wilson[11,12,13] & Jeffrey C. Kwong [1,2,3,4,14,15] ✉

We estimated the effectiveness of booster doses of monovalent mRNA COVID-19 vaccines against Omicron-associated severe outcomes among adults in Ontario, Canada. We used a test-negative design to estimate vaccine effectiveness (VE) against hospitalization or death among SARS-CoV-2-tested adults aged ≥50 years from January 2 to October 1, 2022, stratified by age and time since vaccination. We also compared VE during BA.1/BA.2 and BA.4/BA.5 sublineage predominance. We included 11,160 cases and 62,880 tests for test-negative controls. Depending on the age group, compared to unvaccinated adults, VE was 91–98% 7–59 days after a third dose, waned to 76–87% after ≥240 days, was restored to 92–97% 7–59 days after a fourth dose, and waned to 86–89% after ≥120 days. VE was lower and declined faster during BA.4/BA.5 versus BA.1/BA.2 predominance, particularly after ≥120 days. Here we show that booster doses of monovalent mRNA COVID-19 vaccines restored strong protection against severe outcomes for at least 3 months after vaccination. Across the entire study period, protection declined slightly over time, but waned more during BA.4/BA.5 predominance.

COVID-19 vaccines first became available in Ontario, Canada in December 2020. Due to concerns about waning protection from the primary series and the emergence of more transmissible SARS-CoV-2 variants, third doses (first boosters) were offered to high-risk groups, including community-dwelling adults aged ≥70 years in November 2021[1]. With the emergence of Omicron, the most transmissible and immune-evasive variant to date[2], third dose eligibility was expanded to all adults in December 2021[3]. Ontario began offering fourth doses (second boosters) to adults aged ≥60 years in April 2022[4], and to all adults in July 2022[5]. Booster dose policies differed for residents of long-term care facilities[6]. Bivalent COVID-19 vaccines were introduced to Canadian vaccination programs starting in September 2022 and are now preferred[7,8], but monovalent vaccines are still authorized for use as boosters and are the products that have been most commonly received to date.

[1]Public Health Ontario, Toronto, ON, Canada. [2]ICES, Toronto, ON, Canada. [3]Dalla Lana School of Public Health, University of Toronto, Toronto, ON, Canada. [4]Centre for Vaccine Preventable Diseases, University of Toronto, Toronto, ON, Canada. [5]Institute of Health Policy, Management and Evaluation, University of Toronto, Toronto, ON, Canada. [6]School of Epidemiology and Public Health, University of Ottawa, Ottawa, ON, Canada. [7]Children's Hospital of Eastern Ontario Research Institute, Ottawa, ON, Canada. [8]Department of Laboratory Medicine and Pathobiology, University of Toronto, Toronto, ON, Canada. [9]Women's College Hospital, Toronto, ON, Canada. [10]Leslie Dan Faculty of Pharmacy, University of Toronto, Toronto, ON, Canada. [11]Department of Medicine, University of Ottawa, Ottawa, ON, Canada. [12]Ottawa Hospital Research Institute, Ottawa, ON, Canada. [13]Bruyere Research Institute, Ottawa, ON, Canada. [14]Department of Family and Community Medicine, University of Toronto, Toronto, ON, Canada. [15]University Health Network, Toronto, ON, Canada. ✉e-mail: jeff.kwong@utoronto.ca

**Table 1 | Descriptive characteristics of community-dwelling adults aged ≥50 years tested for SARS-CoV-2 and with severe outcomes between January 2, 2022 and October 1, 2022 in Ontario, Canada, comparing Omicron-associated severe outcome cases to SARS-CoV-2 negative controls, stratified by age group**

| Characteristics | Age 50–59 years | | | Age 60–69 years | | | Age 70–79 years | | | Age 80+ years | | |
|---|---|---|---|---|---|---|---|---|---|---|---|---|
| | SARS-CoV-2 negative controls, n (%)[a] | Omicron cases, n (%) | SD[b] | SARS-CoV-2 negative controls, n (%)[a] | Omicron cases, n (%) | SD[b] | SARS-CoV-2 negative controls, n (%)[a] | Omicron cases, n (%) | SD[b] | SARS-CoV-2 negative controls, n (%)[a] | Omicron cases, n (%) | SD[b] |
| Total | 25,002 | 1136 | | 16,139 | 1926 | | 11,585 | 2975 | | 10,154 | 5.123 | |
| Age (years), mean (standard deviation) | 54.45 ± 2.91 | 55.15 ± 2.86 | 0.24 | 63.95 ± 2.85 | 64.78 ± 2.90 | 0.29 | 74.20 ± 2.81 | 74.79 ± 2.82 | 0.21 | 86.27 ± 4.79 | 86.96 ± 4.93 | 0.14 |
| Male sex | 7370 (29.5%) | 659 (58.0%) | 0.60 | 6,396 (39.6%) | 1,099 (57.1%) | 0.35 | 5,387 (46.5%) | 1,715 (57.6%) | 0.22 | 4,341 (42.8%) | 2,858 (55.8%) | 0.26 |
| **Public health unit region** | | | | | | | | | | | | |
| Central East | 2232 (8.9%) | 62 (5.5%) | 0.13 | 1227 (7.6%) | 118 (6.1%) | 0.06 | 648 (5.6%) | 199 (6.7%) | 0.05 | 387 (3.8%) | 312 (6.1%) | 0.11 |
| Central West | 3794 (15.2%) | 224 (19.7%) | 0.12 | 2408 (14.9%) | 417 (21.7%) | 0.17 | 1815 (15.7%) | 593 (19.9%) | 0.11 | 1609 (15.8%) | 950 (18.5%) | 0.07 |
| Durham | 1552 (6.2%) | 48 (4.2%) | 0.09 | 829 (5.1%) | 52 (2.7%) | 0.13 | 375 (3.2%) | 88 (3.0%) | 0.02 | 133 (1.3%) | 121 (2.4%) | 0.08 |
| Eastern | 1356 (5.4%) | 105 (9.2%) | 0.15 | 621 (3.8%) | 168 (8.7%) | 0.20 | 347 (3.0%) | 265 (8.9%) | 0.25 | 310 (3.1%) | 394 (7.7%) | 0.21 |
| North | 4696 (18.8%) | 126 (11.1%) | 0.22 | 3428 (21.2%) | 183 (9.5%) | 0.33 | 2533 (21.9%) | 249 (8.4%) | 0.38 | 2180 (21.5%) | 378 (7.4%) | 0.41 |
| Ottawa | 599 (2.4%) | 57 (5.0%) | 0.14 | 274 (1.7%) | 83 (4.3%) | 0.15 | 166 (1.4%) | 147 (4.9%) | 0.20 | 187 (1.8%) | 240 (4.7%) | 0.16 |
| Peel | 2266 (9.1%) | 125 (11.0%) | 0.06 | 1670 (10.3%) | 212 (11.0%) | 0.02 | 1486 (12.8%) | 359 (12.1%) | 0.02 | 1613 (15.9%) | 579 (11.3%) | 0.13 |
| South West | 3961 (15.8%) | 165 (14.5%) | 0.04 | 2980 (18.5%) | 271 (14.1%) | 0.12 | 2425 (20.9%) | 409 (13.7%) | 0.19 | 2288 (22.5%) | 604 (11.8%) | 0.29 |
| Toronto | 3122 (12.5%) | 143 (12.6%) | 0.00 | 1888 (11.7%) | 286 (14.8%) | 0.09 | 1384 (11.9%) | 442 (14.9%) | 0.09 | 1234 (12.2%) | 1098 (21.4%) | 0.25 |
| York | 1329 (5.3%) | 74 (6.5%) | 0.05 | 760 (4.7%) | 129 (6.7%) | 0.09 | 382 (3.3%) | 210 (7.1%) | 0.17 | 169 (1.7%) | 432 (8.4%) | 0.31 |
| Missing | 95 (0.4%) | 7 (0.6%) | 0.03 | 54 (0.3%) | 7 (0.4%) | 0.03 | 24 (0.2%) | 14 (0.5%) | 0.05 | 44 (0.4%) | 15 (0.3%) | 0.02 |
| **Household income quintile** | | | | | | | | | | | | |
| 1 (lowest) | 4645 (18.6%) | 337 (29.7%) | 0.26 | 3552 (22.0%) | 577 (30.0%) | 0.18 | 2574 (22.2%) | 824 (27.7%) | 0.13 | 2368 (23.3%) | 1248 (24.4%) | 0.02 |
| 2 | 4841 (19.4%) | 221 (19.5%) | 0.00 | 3331 (20.6%) | 426 (22.1%) | 0.05 | 2415 (20.8%) | 663 (22.3%) | 0.04 | 2291 (22.6%) | 1170 (22.8%) | 0.01 |
| 3 | 4955 (19.8%) | 216 (19.0%) | 0.02 | 3001 (18.6%) | 336 (17.4%) | 0.02 | 2215 (19.1%) | 552 (18.6%) | 0.01 | 1963 (19.3%) | 1024 (20.0%) | 0.02 |
| 4 | 5091 (20.4%) | 215 (18.9%) | 0.04 | 3053 (18.9%) | 332 (17.2%) | 0.00 | 2174 (18.8%) | 498 (16.7%) | 0.05 | 1797 (17.7%) | 885 (17.3%) | 0.01 |
| 5 (highest) | 4595 (18.4%) | 307 (27.0%) | 0.21 | 3152 (19.5%) | 483 (25.1%) | 0.24 | 2168 (18.7%) | 714 (24.0%) | 0.13 | 1687 (16.6%) | 785 (15.3%) | 0.04 |
| Missing | 63 (0.3%) | 3 (0.3%) | 0.02 | 50 (0.3%) | 5 (0.3%) | 0.00 | 39 (0.3%) | 19 (0.6%) | 0.01 | 48 (0.5%) | 11 (0.2%) | 0.04 |
| **Essential workers quintile** | | | | | | | | | | | | |
| 1 (0%–32.5%) | 3941 (15.8%) | 110 (9.7%) | 0.18 | 2266 (14.0%) | 209 (10.9%) | 0.18 | 1782 (15.4%) | 382 (12.8%) | 0.10 | 1699 (16.7%) | 865 (16.9%) | 0.00 |
| 2 (32.5%–42.3%) | 5669 (22.7%) | 236 (20.8%) | 0.05 | 3354 (20.8%) | 361 (18.7%) | 0.05 | 2368 (20.4%) | 581 (19.5%) | 0.02 | 2230 (22.0%) | 1085 (21.2%) | 0.02 |
| 3 (42.3%–49.8%) | 5429 (21.7%) | 236 (20.8%) | 0.02 | 3535 (21.9%) | 420 (21.8%) | 0.00 | 2503 (21.6%) | 621 (20.9%) | 0.02 | 2264 (22.3%) | 1087 (21.2%) | 0.03 |
| 4 (50.0%–57.5%) | 5157 (20.6%) | 235 (20.7%) | 0.00 | 3440 (21.3%) | 437 (22.7%) | 0.03 | 2429 (21.0%) | 654 (22.0%) | 0.02 | 1938 (19.1%) | 1112 (21.7%) | 0.07 |
| 5 (57.5%–100%) | 5407 (21.6%) | 144 (12.7%) | 0.24 | 3403 (21.1%) | 483 (25.1%) | 0.09 | 2437 (21.0%) | 714 (24.0%) | 0.07 | 1962 (19.3%) | 953 (18.6%) | 0.02 |
| Missing | 211 (0.8%) | 12 (1.1%) | 0.02 | 141 (0.9%) | 16 (0.8%) | 0.01 | 66 (0.6%) | 23 (0.8%) | 0.02 | 61 (0.6%) | 21 (0.4%) | 0.03 |
| **Persons per dwelling quintile** | | | | | | | | | | | | |
| 1 (0–2.1) | 4518 (18.1%) | 263 (23.2%) | 0.13 | 3647 (22.6%) | 489 (25.4%) | 0.13 | 3131 (27.0%) | 805 (27.1%) | 0.00 | 3268 (32.2%) | 1362 (26.6%) | 0.12 |
| 2 (2.2–2.4) | 5220 (20.9%) | 235 (20.7%) | 0.00 | 3682 (22.8%) | 396 (20.6%) | 0.00 | 2733 (23.6%) | 577 (19.4%) | 0.10 | 2344 (23.1%) | 958 (18.7%) | 0.11 |
| 3 (2.5–2.6) | 3480 (13.9%) | 168 (14.8%) | 0.02 | 2182 (13.5%) | 281 (14.6%) | 0.02 | 1505 (13.0%) | 411 (13.8%) | 0.02 | 1345 (13.2%) | 732 (14.3%) | 0.03 |
| 4 (2.7–3.0) | 5747 (23.0%) | 225 (19.8%) | 0.08 | 3269 (20.3%) | 359 (18.6%) | 0.08 | 2308 (19.9%) | 561 (18.9%) | 0.03 | 1822 (17.9%) | 1084 (21.2%) | 0.08 |
| 5 (3.1–5.7) | 5824 (23.3%) | 233 (20.5%) | 0.07 | 3203 (19.8%) | 382 (19.8%) | 0.07 | 1844 (15.9%) | 595 (20.0%) | 0.11 | 1303 (12.8%) | 962 (18.8%) | 0.16 |
| Missing | 213 (0.9%) | 12 (1.1%) | 0.02 | 156 (1.0%) | 19 (1.0%) | 0.02 | 64 (0.6%) | 26 (0.9%) | 0.04 | 72 (0.7%) | 25 (0.5%) | 0.03 |

**Table 1 (continued) | Descriptive characteristics of community-dwelling adults aged ≥50 years tested for SARS-CoV-2 and with severe outcomes between January 2, 2022 and October 1, 2022 in Ontario, Canada, comparing Omicron-associated severe outcome cases to SARS-CoV-2 negative controls, stratified by age group**

| | Age 50–59 years | | | Age 60–69 years | | | Age 70–79 years | | | Age 80+ years | | |
|---|---|---|---|---|---|---|---|---|---|---|---|---|
| | SARS-CoV-2 negative controls, n (%)[a] | Omicron cases, n (%) | SD[b] | SARS-CoV-2 negative controls, n (%)[a] | Omicron cases, n (%) | SD[b] | SARS-CoV-2 negative controls, n (%)[a] | Omicron cases, n (%) | SD[b] | SARS-CoV-2 negative controls, n (%)[a] | Omicron cases, n (%) | SD[b] |
| **Self-identified visible minority quintile** | | | | | | | | | | | | |
| 1 (0.0%–2.2%) | 5864 (23.5%) | 247 (21.7%) | 0.04 | 4054 (25.1%) | 392 (20.4%) | 0.11 | 2811 (24.3%) | 605 (20.3%) | 0.09 | 2126 (20.9%) | 865 (16.9%) | 0.10 |
| 2 (2.2%–7.5%) | 5445 (21.8%) | 181 (15.9%) | 0.15 | 3624 (22.5%) | 345 (17.9%) | 0.11 | 2675 (23.1%) | 548 (18.4%) | 0.12 | 2305 (22.7%) | 902 (17.6%) | 0.13 |
| 3 (7.5%–18.7%) | 4387 (17.5%) | 192 (16.9%) | 0.02 | 2771 (17.2%) | 356 (18.5%) | 0.03 | 2039 (17.6%) | 514 (17.3%) | 0.01 | 2150 (21.2%) | 955 (18.6%) | 0.06 |
| 4 (18.7%–43.5%) | 4174 (16.7%) | 231 (20.3%) | 0.09 | 2488 (15.4%) | 347 (18.0%) | 0.07 | 1963 (16.9%) | 555 (18.7%) | 0.04 | 1894 (18.7%) | 1069 (20.9%) | 0.06 |
| 5 (43.5%–100%) | 4921 (19.7%) | 273 (24.0%) | 0.11 | 3061 (19.0%) | 470 (24.4%) | 0.13 | 2031 (17.5%) | 731 (24.6%) | 0.17 | 1618 (15.9%) | 1311 (25.6%) | 0.24 |
| Missing | 211 (0.8%) | 12 (1.1%) | 0.02 | 141 (0.9%) | 16 (0.8%) | 0.00 | 66 (0.6%) | 22 (0.7%) | 0.02 | 61 (0.6%) | 21 (0.4%) | 0.03 |
| Receipt of 2019–2020 and/or 2020–2021 influenza vaccination | 8842 (35.4%) | 311 (27.4%) | 0.17 | 8064 (50.0%) | 777 (40.3%) | 0.19 | 8094 (69.9%) | 1594 (53.6%) | 0.34 | 7573 (74.6%) | 3249 (63.4%) | 0.24 |
| Prior positive SARS-CoV-2 test | 1674 (6.7%) | 29 (2.6%) | 0.20 | 789 (4.9%) | 21 (1.1%) | 0.22 | 345 (3.0%) | 35 (1.2%) | 0.13 | 248 (2.4%) | 43 (0.8%) | 0.13 |
| **Number of SARS-CoV-2 tests within 3 months prior to December 14, 2020** | | | | | | | | | | | | |
| 0 | 15,958 (63.8%) | 957 (84.2%) | 0.48 | 11,629 (72.1%) | 1660 (86.2%) | 0.35 | 9622 (83.1%) | 2662 (89.5%) | 0.19 | 8676 (85.4%) | 4596 (89.7%) | 0.13 |
| 1 | 4364 (17.5%) | 117 (10.3%) | 0.21 | 2337 (14.5%) | 187 (9.7%) | 0.15 | 1341 (11.6%) | 225 (7.6%) | 0.14 | 953 (9.4%) | 361 (7.0%) | 0.09 |
| ≥2 | 4680 (18.7%) | 62 (5.5%) | 0.42 | 2173 (13.5%) | 79 (4.1%) | 0.34 | 622 (5.4%) | 88 (3.0%) | 0.12 | 525 (5.2%) | 166 (3.2%) | 0.10 |
| Any comorbidity | 14,709 (58.8%) | 893 (78.6%) | 0.44 | 12,236 (75.8%) | 1665 (86.4%) | 0.27 | 10,362 (89.4%) | 2791 (93.8%) | 0.16 | 9826 (96.8%) | 4995 (97.5%) | 0.04 |
| **Receipt of home care services** | | | | | | | | | | | | |
| None | 24,611 (98.4%) | 1058 (93.1%) | 0.27 | 15,486 (96.0%) | 1791 (93.0%) | 0.13 | 10,683 (92.2%) | 2726 (91.6%) | 0.02 | 8861 (87.3%) | 4537 (88.6%) | 0.04 |
| Short stay | 279 (1.1%) | 41 (3.6%) | 0.16 | 402 (2.5%) | 64 (3.3%) | 0.05 | 459 (4.0%) | 104 (3.5%) | 0.02 | 512 (5.0%) | 162 (3.2%) | 0.09 |
| Long stay | 96 (0.4%) | 32–36 (2.8–3.2%) | 0.20 | 217 (1.3%) | 60 (3.1%) | 0.12 | 394 (3.4%) | 135 (4.5%) | 0.06 | 734 (7.2%) | 401 (7.8%) | 0.02 |
| Palliative | 16 (0.1%) | ≤5 (≤0.4%) | 0.05 | 34 (0.2%) | 11 (0.6%) | 0.06 | 49 (0.4%) | 10 (0.3%) | 0.01 | 47 (0.5%) | 23 (0.4%) | 0.00 |
| Unvaccinated[c] | 944 (3.8%) | 416 (36.6%) | 0.90 | 840 (5.2%) | 695 (36.1%) | 0.83 | 572 (4.9%) | 880 (29.6%) | 0.69 | 538 (5.3%) | 1136 (22.2%) | 0.51 |
| **Time since second dose[c]** | | | | | | | | | | | | |
| 7–59 days | 119 (0.5%) | 10 (0.9%) | 0.05 | 60 (0.4%) | 10 (0.5%) | 0.02 | 25 (0.2%) | 6 (0.2%) | 0.00 | 27 (0.3%) | 10 (0.2%) | 0.01 |
| 60–119 days | 396 (1.6%) | 37 (3.3%) | 0.11 | 149 (0.9%) | 27 (1.4%) | 0.04 | 82 (0.7%) | 27 (0.9%) | 0.02 | 32 (0.3%) | 34 (0.7%) | 0.05 |
| 120–179 days | 1066 (4.3%) | 65 (5.7%) | 0.07 | 544 (3.4%) | 107 (5.6%) | 0.11 | 205 (1.8%) | 112 (3.8%) | 0.12 | 135 (1.3%) | 125 (2.4%) | 0.08 |
| 180–239 days | 2246 (9.0%) | 136 (12.0%) | 0.10 | 1,230 (7.6%) | 206 (10.7%) | 0.11 | 538 (4.6%) | 332 (11.2%) | 0.24 | 423 (4.2%) | 542 (10.6%) | 0.25 |
| 240–299 days | 848 (3.4%) | 55 (4.8%) | 0.07 | 524 (3.2%) | 108 (5.6%) | 0.11 | 357 (3.1%) | 133 (4.5%) | 0.07 | 290 (2.9%) | 180 (3.5%) | 0.04 |
| ≥300 days | 1,215 (4.9%) | 86 (7.6%) | 0.11 | 778 (4.8%) | 139 (7.2%) | 0.10 | 447 (3.9%) | 163 (5.5%) | 0.08 | 471 (4.6%) | 363 (7.1%) | 0.10 |
| **Time since third dose[c]** | | | | | | | | | | | | |
| 0–6 days | 402 (1.6%) | 8 (0.7%) | 0.08 | 233 (1.4%) | 14 (0.7%) | 0.07 | 109 (0.9%) | 30 (1.0%) | 0.01 | 65 (0.6%) | 49 (1.0%) | 0.04 |
| 7–59 days | 5681 (22.7%) | 68 (6.0%) | 0.49 | 3650 (22.6%) | 94 (4.9%) | 0.53 | 2101 (18.1%) | 182 (6.1%) | 0.37 | 1496 (14.7%) | 299 (5.8%) | 0.30 |
| 60–119 days | 4830 (19.3%) | 80 (7.0%) | 0.37 | 2892 (17.9%) | 145 (7.5%) | 0.32 | 1994 (17.2%) | 243 (8.2%) | 0.27 | 1902 (18.7%) | 503 (9.8%) | 0.26 |
| 120–179 days | 3583 (14.3%) | 50 (4.4%) | 0.35 | 2104 (13.0%) | 117 (6.1%) | 0.24 | 1699 (14.7%) | 246 (8.3%) | 0.20 | 1400 (13.8%) | 509 (9.9%) | 0.12 |
| 180–239 days | 2086 (8.3%) | 73 (6.4%) | 0.07 | 1116 (6.9%) | 128 (6.6%) | 0.01 | 816 (7.0%) | 226 (7.6%) | 0.02 | 685 (6.7%) | 448 (8.7%) | 0.07 |
| ≥240 days | 1040 (4.2%) | 40 (3.5%) | 0.03 | 479 (3.0%) | 46 (2.4%) | 0.04 | 343 (3.0%) | 109 (3.7%) | 0.04 | 324 (3.2%) | 252 (4.9%) | 0.09 |
| **Time since fourth dose[c]** | | | | | | | | | | | | |
| 0–6 days | 32 (0.1%) | ≤5 (≤0.4%)[d] | 0.01 | 95 (0.6%) | ≤5 (≤0.3%)[d] | 0.01 | 103 (0.9%) | 10 (0.3%) | 0.07 | 83 (0.8%) | 29 (0.6%) | 0.03 |

**Table 1 (continued) | Descriptive characteristics of community-dwelling adults aged ≥50 years tested for SARS-CoV-2 and with severe outcomes between January 2, 2022 and October 1, 2022 in Ontario, Canada, comparing Omicron-associated severe outcome cases to SARS-CoV-2 negative controls, stratified by age group**

| | Age 50–59 years | | | Age 60–69 years | | | Age 70–79 years | | | Age 80+ years | | |
|---|---|---|---|---|---|---|---|---|---|---|---|---|
| | SARS-CoV-2 negative controls, n (%)a | Omicron cases, n (%) | SDb | SARS-CoV-2 negative controls, n (%)a | Omicron cases, n (%) | SDb | SARS-CoV-2 negative controls, n (%)a | Omicron cases, n (%) | SDb | SARS-CoV-2 negative controls, n (%)a | Omicron cases, n (%) | SDb |
| 7–59 days | 357 (1.4%) | ≤5 (≤0.4%)d | 0.11 | 730 (4.5%) | 28 (1.5%) | 0.18 | 979 (8.5%) | 85 (2.9%) | 0.24 | 905 (8.9%) | 152 (3.0%) | 0.25 |
| 60–119 days | 119 (0.5%) | ≤5 (≤0.4%)d | 0.03 | 511 (3.2%) | 40 (2.1%) | 0.07 | 837 (7.2%) | 106 (3.6%) | 0.16 | 835 (8.2%) | 255 (5.0%) | 0.13 |
| ≥120 days | 38 (0.2%) | ≤5 (≤0.4%)d | 0.02 | 204 (1.3%) | 17–21 (0.9–1.1%)d | 0.02 | 378 (3.3%) | 85 (2.9%) | 0.02 | 543 (5.3%) | 237 (4.6%) | 0.03 |

Note, not unique by person; individuals may be included more than once.
aProportion reported, unless stated otherwise.
bSD = standardized difference. Standardized differences of >0.10 are considered clinically relevant. Comparing Omicron cases to test-negative controls.
cSum of all rows (unvaccinated and vaccinated) equals 100%.
dDue to institutional privacy policies, any cells ≤5 (except for missing values) must be suppressed and ranges must be provided for complementary cells to prevent back calculation.

Various Omicron sublineages have circulated during 2022, with BA.1 and BA.2 predominating until June, and BA.4 and BA.5 predominating subsequently. The seroprevalence of prior SARS-CoV-2 infection for the overall population increased substantially in Ontario during the Omicron period, from 6% in early January 2022 to approximately 50% by early July 2022 and 63% by early October 2022[9]. Across Canada, seroprevalence is lower when restricted to adults aged ≥60 years (50% by early October 2022)[9].

Due to increased transmissibility and immune evasion of emerging Omicron sublineages, more evidence is needed on the long-term effectiveness of booster doses of monovalent mRNA vaccines among older adults to inform planning for subsequent boosters and future shifts in vaccine development. Thus, we sought to estimate vaccine effectiveness (VE) of 2, 3, and 4 doses compared to unvaccinated subjects, and marginal effectiveness of 3 or 4 doses compared to 2 doses, in preventing severe outcomes (hospitalization or death) among community-dwelling adults aged ≥50 years during an Omicron-dominant period. We estimated marginal effectiveness due to concerns about differences between unvaccinated and vaccinated populations. We also sought to determine how VE varied during periods of BA.1/BA.2 versus BA.4/BA.5 predominance.

## Results

We included 11,160 Omicron-associated severe outcomes and 62,880 symptomatic test-negative controls (among 53,369 individuals). Across all study age groups, more cases than controls were male and fewer were unvaccinated (Table 1). Among the 50–59 and 60–69 years age groups, more cases had at least one comorbid condition and were from areas with the lowest incomes. Across all age groups, compared to unvaccinated subjects, more vaccinated subjects had previously received influenza vaccines (Supplementary Tables 2–5). This difference was also seen when comparing subjects with 2 versus 3 or 4 doses (Supplementary Tables 6–9). Among those aged 70–79 years, 36% and 31% of subjects who received a third and fourth dose, respectively, received the mRNA-1273 vaccine. Among subjects aged ≥80 years, 34% and 37% of third and fourth dose recipients received mRNA-1273, respectively.

### Vaccine effectiveness and marginal effectiveness

Compared to unvaccinated subjects, VE against severe disease increased shortly after receipt of booster doses but subsequently declined over time (Fig. 1, Supplementary Tables 10–11). For example, among subjects aged 70–79 years, VE decreased from: 84% (95% CI, 57–94%) 7–59 days after a second dose to 71% (95% CI, 63–78%) after ≥300 days; 96% (95% CI, 95–97%) 7–59 days after a third dose to 79% (95% CI, 71–85%) after ≥240 days; and 93% (95% CI, 91–95%) 7–59 days after a fourth dose to 89% (95% CI, 84–92%) after ≥120 days. The decline in VE after a third dose appeared to plateau after 180 days. VE was generally lower with increasing age.

Marginal effectiveness peaked 7–59 days after third and fourth doses and declined over time. Once again, among subjects aged 70–79 years, compared to 2 doses (median 218 days since a second dose), the marginal effectiveness of a third dose decreased from 83% (95% CI, 79–86%) 7–59 days after a third dose to 45% (95% CI, 24–59%) after ≥240 days and the marginal effectiveness of a fourth dose decreased from 80% (95% CI, 74–85%) 7–59 days after a fourth dose to 69% (95% CI, 57–78%) after ≥120 days (Fig. 2, Supplementary Table 12).

### Additional analyses

VE estimates were lower during the BA.4/BA.5-predominant period compared to the BA.1/BA.2-predominant period, with differences widening as time since vaccination increased (Fig. 3, Supplementary Table 13). For example, among subjects aged 70–79 years, VE 7–59 days after a third dose was 96% (95% CI, 96–97%) (median 34 days since a third dose) during the BA.1/BA.2-predominant period compared to

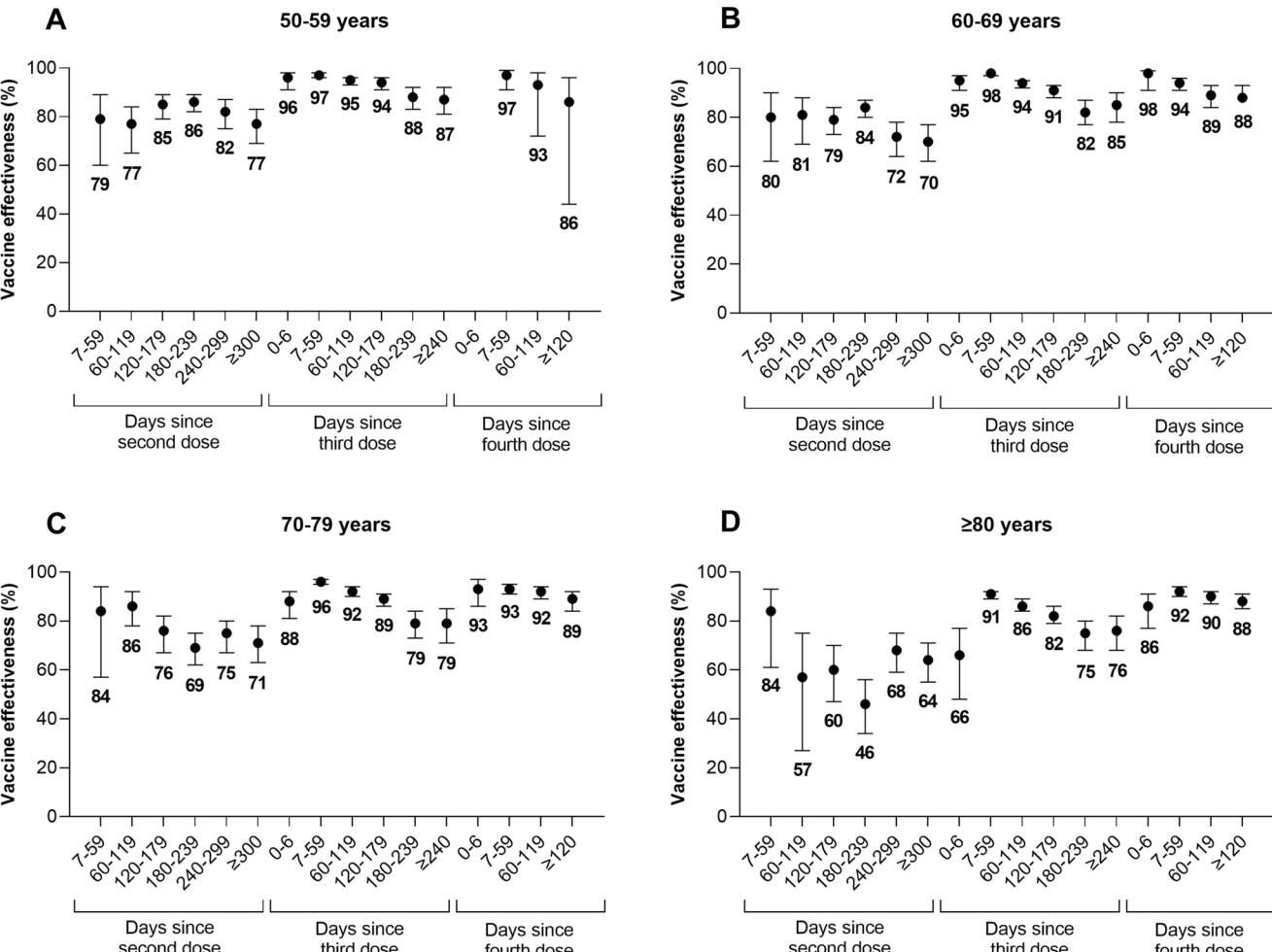

**Fig. 1 | Vaccine effectiveness and 95% confidence intervals by time since vaccination.** Vaccine effectiveness (presented as proportions out of 100 percentage points) and 95% confidence intervals of 2, 3, and 4 doses of monovalent mRNA COVID-19 vaccines against Omicron-associated severe outcomes by time since vaccination among community-dwelling adults aged (**A**) 50–59 years, (**B**) 60–69 years, (**C**) 70–79 years, and (**D**) ≥ 80 years in Ontario, Canada, compared to unvaccinated adults (Note: Estimates were not reported if they were unstable [i.e., 95% confidence interval width exceeded 100 percentage points]). Please see Supplementary Table 10 for all estimates.

86% (95% CI, 37–97%) (median 41 days since a third dose) during the BA.4/BA.5-predominant period (*p* = 0.08 for the between-period interaction), whereas VE 180–239 days after a third dose was 91% (95% CI, 85–95%) (median 189 days since a third dose) during the BA.1/BA.2-predominant period compared to 59% (95% CI, 44–70%) (median 214 days since a third dose) during the BA.4/BA.5-predominant period (*p* < 0.001 for the between-period interaction).

When Paxlovid recipients were removed from the analysis, VE estimates were nearly identical to those from the main analysis (Supplementary Table 14).

## Discussion

Among community-dwelling adults aged ≥50 years in Ontario, VE against Omicron-associated severe outcomes increased with booster doses of monovalent mRNA COVID-19 vaccines, but protection waned over time after each dose. Third doses continued to provide strong protection (85–87%) against severe outcomes among subjects aged 50–69 years even 8 months after vaccination, but lower protection (76–79%) among those aged ≥70 years. Fourth doses restored waning of protection from third doses and continued to provide strong protection (86–89%) 4 months after vaccination for all age groups. However, VE in the BA.4/BA.5-predominant period was lower than during

the BA.1/BA.2-predominant period across the same time intervals after vaccination, especially with increasing time since vaccination.

Comparisons with other jurisdictions are challenging due to heterogeneity in study designs, population characteristics, outcomes and exposures, vaccines, and observation periods. Our fourth dose VE estimates were slightly higher than those observed in the United States, where fourth dose VE against hospitalizations was 80% (95%CI, 71–85%) after ≥7 days among adults aged ≥50 years[10]. Studies from Israel found that waning of protection against severe outcomes was significantly slower than against infection and that marginal effectiveness of booster doses against infection waned faster after fourth doses compared to third doses[11-13]. They were unable to determine if trends were similar for severe outcomes due to the short follow-up period. In our study, the waning of protection against severe outcomes observed ≥120 days after a fourth dose was comparable to that seen 120–179 days after a third dose. Although differences in timing of vaccination within those time periods may influence VE estimates, the median time since vaccination was 141–145 days (depending on the age group) for the 120–179 days post third dose group and 140–147 days for the ≥120 days post fourth dose group, suggesting that waning of protection after a fourth dose may follow a similar trajectory as after a third dose.

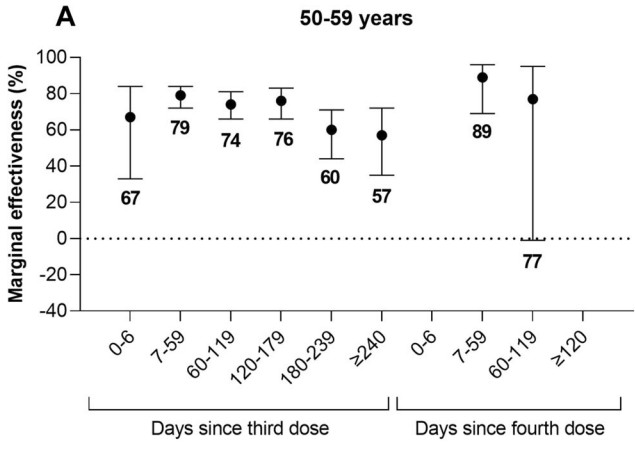

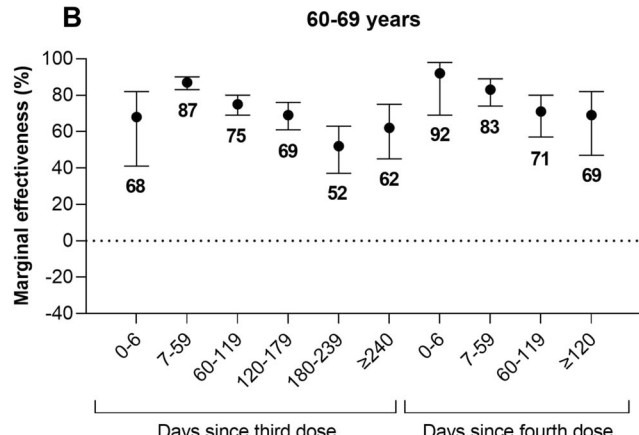

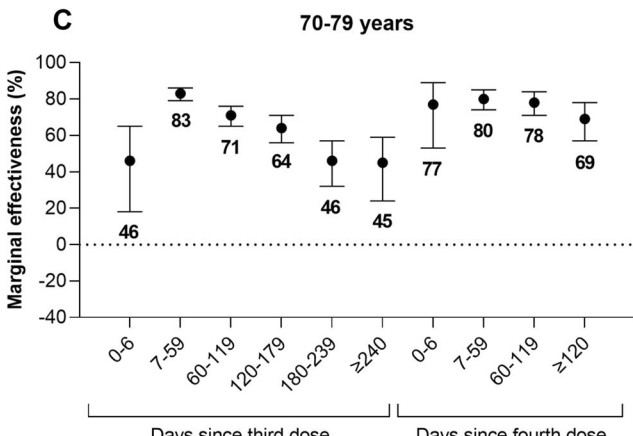

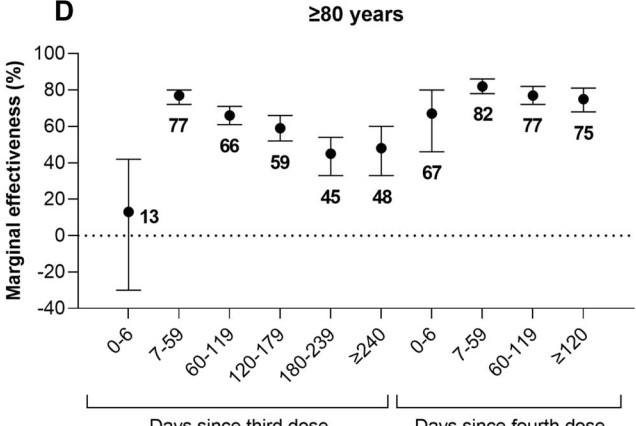

**Fig. 2 | Marginal effectiveness and 95% confidence intervals by time since vaccination.** Marginal effectiveness (presented as proportions out of 100 percentage points) and 95% confidence intervals of a third and fourth dose of monovalent mRNA COVID-19 vaccines against Omicron-associated severe outcomes by time since vaccination among community-dwelling adults aged (**A**) 50–59 years, (**B**) 60–69 years, (**C**) 70–79 years, and (**D**) ≥80 years in Ontario, Canada, compared to adults who received 2 doses. (Note: Estimates were not reported if they were unstable [i.e., 95% confidence interval width exceeded 100 percentage points]). Please see Supplementary Table 12 for all estimates.

Available evidence on VE of monovalent COVID-19 vaccines against severe outcomes among adults aged ≥18 years caused by the BA.4/BA.5 Omicron sublineages varies. In the UK, compared to a second dose, marginal effectiveness of a third or fourth dose against BA.4/BA.5- versus BA.2-related hospitalizations was similar using the same time intervals since vaccination[14]. Similarly, VE of a third dose against hospitalizations was comparable between BA.1/BA.2-predominant and BA.4/BA.5-predominant periods in South Africa[15]. Conversely, in Portugal, 3-dose protection against severe outcomes was lower among BA.5 versus BA.2 cases[16]. A study among Kaiser Permanente members found that VE of third and fourth doses against BA.4/BA.5-related hospitalizations was lower compared to BA.1/BA.2-related hospitalizations, whereas another study among individuals admitted to IVY Network hospitals saw this difference for a third dose but not for 2 or 4 doses[17,18]. The IVY Network study reported 3-dose VE of 79% (95% CI, 74–84%) and 60% (95% CI, 12–81%), respectively, during the BA.1/BA.2-predominant versus BA.4/BA.5-predominant periods 7–120 days after vaccination[18].

Potential explanations for lower VE during the period of BA.4/BA.5-predominance compared to BA.1/BA.2-predominance include longer intervals between booster dose receipt and outcomes, increased incidence of undocumented infections, and increased BA.4/BA.5 immune evasion[17]. In our study, differences in the median numbers of days since booster receipt between the BA.1/BA.2-predominant

versus BA.4/BA.5-predominant periods were always <30 days, so longer post-booster follow-up during the BA.4/BA.5-predominant period was unlikely to be a major contributor to the large observed differences in VE. VE may be underestimated in the setting of undocumented infections if unvaccinated individuals are more likely to be infected than vaccinated individuals because the former will have infection-induced immunity, and the extent of VE underestimation may increase as prior infections become more prevalent in the population. During the BA.1/BA.2-predominant period, infection-acquired seroprevalence in Ontario in the overall population increased from 6% to 50% and subsequently increased from 50% to 63% during the BA.4/BA.5-predominant period[9]. Seroprevalence was lower among adults aged ≥60 years[9]. However, we noted that VE declined considerably faster as time since vaccination increased during the relatively brief BA.4/BA.5-predominant period (only 3 months) compared to the BA.1/BA.2-predominant period, suggesting that bias from undocumented prior infections is unlikely to account entirely for the differences. Therefore, among these potential explanations, increased immune evasion by BA.4/BA.5 sublineages is likely the largest contributor to these differences in VE.

Based on the evidence to date, the level of protection offered by bivalent vaccines remains unclear. Findings from a phase 2-3 trial suggest that Moderna's bivalent vaccine elicits higher titres of neutralizing antibodies against Omicron sublineages BA.1 and

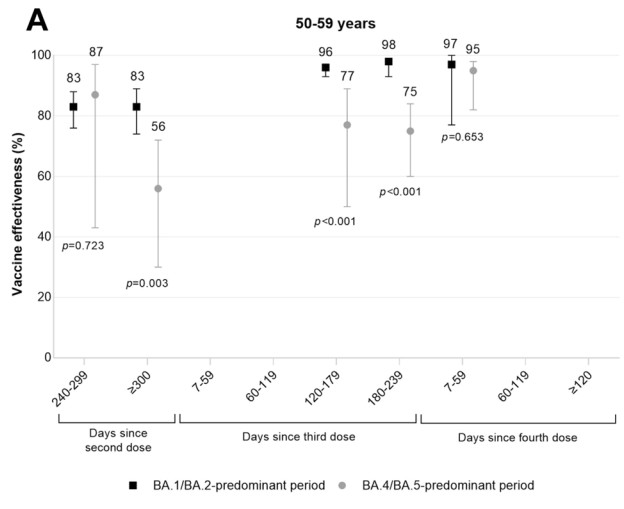

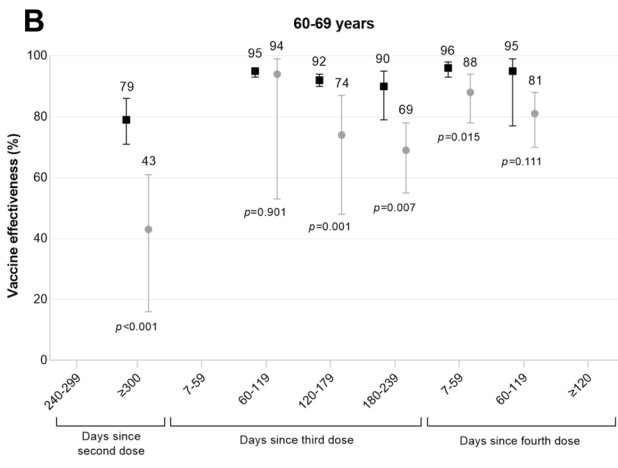

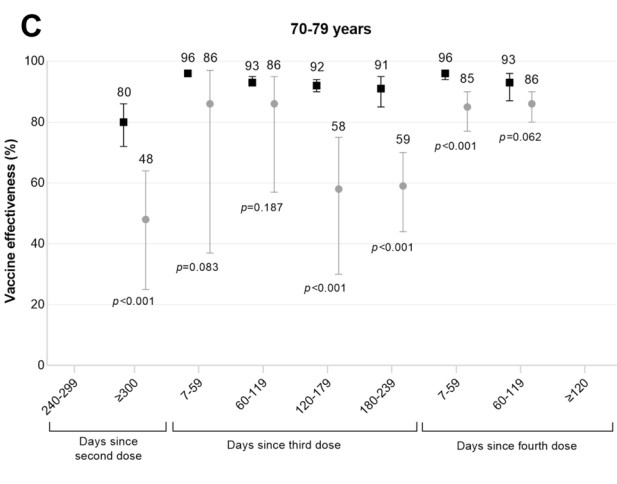

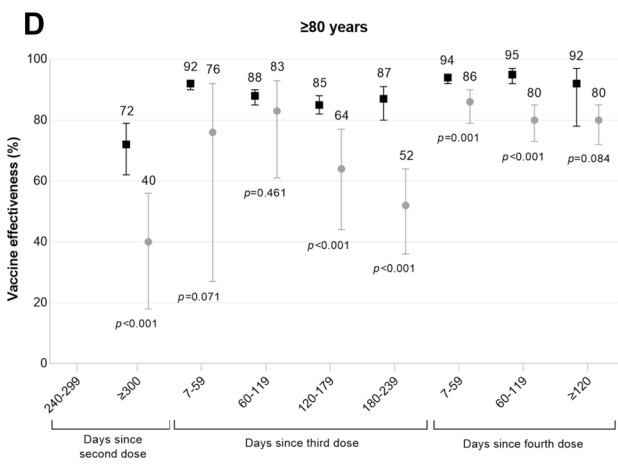

**Fig. 3 | Vaccine effectiveness and 95% confidence intervals, during periods of BA.1/BA.2 (January 2 to July 2, 2022) and BA.4/BA.5 (July 3 to October 1, 2022) predominance, by time since vaccination.** Vaccine effectiveness (presented as proportions out of 100 percentage points) and 95% confidence intervals against Omicron-associated severe outcomes among community-dwelling adults aged (**A**) 50–59 years, (**B**) 60–69 years, (**C**) 70–79 years, and (**D**) ≥80 years in Ontario,
Canada, comparing those who received ≥2 doses of monovalent mRNA COVID-19 vaccines to those who received none, by age and time since vaccination, during periods of BA.1/BA.2 (January 2 to July 2, 2022) and BA.4/BA.5 (July 3 to October 1, 2022) predominance. (Note: Estimates were not reported if they were unstable [i.e., 95% confidence interval width exceeded 100 percentage points] for either period). Please see Supplementary Table 13 for all estimates.

BA.4/BA.5 compared to Moderna's ancestral monovalent vaccine[19]. In contrast, two observational immunogenicity studies found the Pfizer and Moderna bivalent vaccines elicited similar levels of neutralizing antibodies against BA.4/BA.5 as the ancestral monovalent vaccines[20,21].

This study had some limitations. First, data on rapid antigen tests were not available, and this was the main source of testing after December 31, 2021, when eligibility for RT-PCR testing was restricted in Ontario to individuals considered at high risk of acquiring SARS-CoV-2[22]. Thus, while we adjusted for prior SARS-CoV-2 infections documented by RT-PCR, we could not account for prior infections confirmed only by rapid antigen tests. This could bias VE estimates downward or upward depending on whether unvaccinated or vaccinated individuals are more likely to have prior undocumented infections. Second, because whole genome sequencing was not performed on all cases, we were unable to estimate VE against BA.1, BA.2, BA.4, and BA.5 separately but instead combined BA.1 and BA.2 during one period and BA.4 and BA.5 during another period based on when each grouping circulated. Last, there remains the potential for residual confounding since we were limited to the covariates available in the databases used. Unmeasured differences between unvaccinated and

vaccinated individuals may introduce bias, but the consistency between the VE and marginal effectiveness estimates is reassuring. A significant strength of our study is the length of the follow-up period, allowing us to estimate VE ≥ 4 months after fourth doses. Also, unlike most other studies, we stratified our analyses by age group, which provides more refined VE estimates for decision-making.

Our findings suggest that while booster doses of monovalent mRNA COVID-19 vaccines initially restore strong protection against Omicron-associated hospitalizations and death among community-dwelling older adults and then subsequently wane over time, much uncertainty remains. Although protection remained strong 4 months after a fourth dose for all age groups, whether waning increases past this period remains unknown, and combined with the evidence of reduced VE against BA.4/BA.5 sublineages and the possibility that vaccines could be even less effective against newly emerging sublineages such as BQ.1.1 and XBB, subsequent boosters and other measures (e.g., face masks, improved ventilation, filtration of indoor air) may be needed to mitigate the impact of Omicron and future SARS-CoV-2 variants. It will be important to continue monitoring VE given the scarcity of VE data against BA.4/BA.5, newly emerging sublineages, and the introduction of bivalent vaccines.

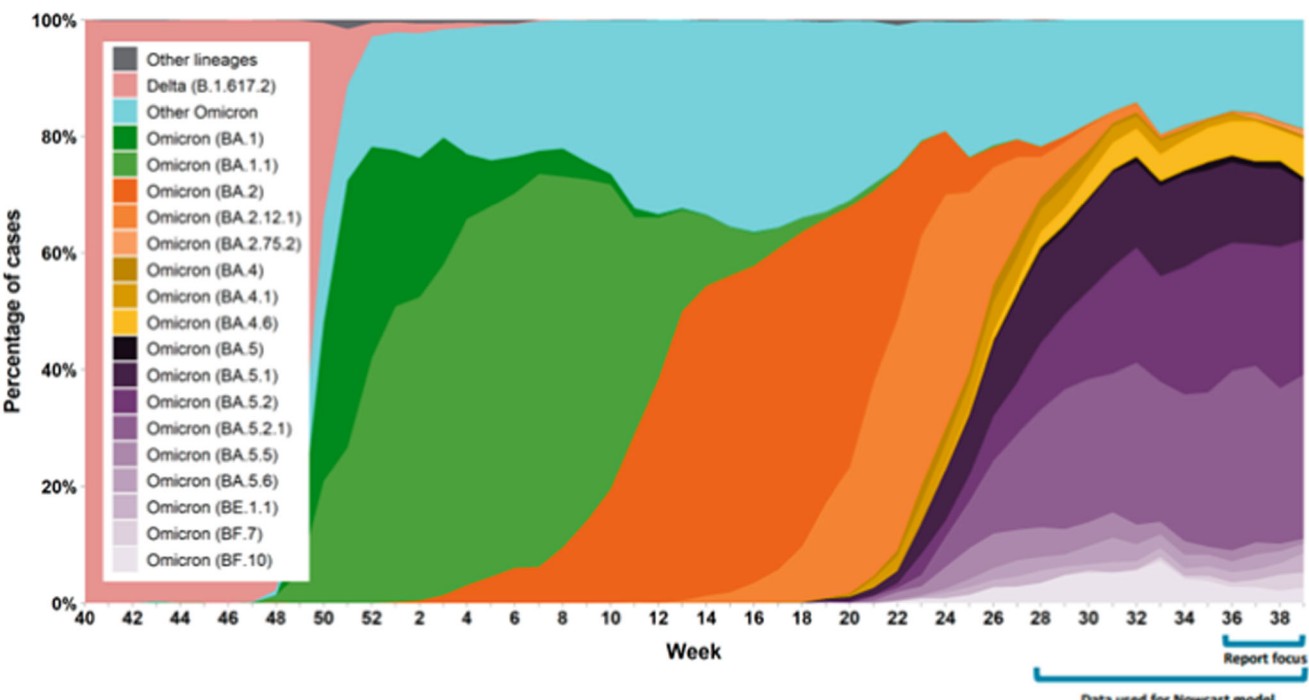

**Fig. 4 | Percentage of COVID-19 cases by the most prevalent lineages and week in Ontario from October 3, 2021 to October 1, 2022.** Percentage of COVID-19 cases by the most prevalent lineages and week, representative surveillance, Ontario, October 3, 2021 to October 1, 2022[27]. Each color represents a different lineage or sublineage of Omicron. Note: results may not be representative of Ontario overall, particularly in earlier weeks. Week was assigned based on earliest date available for a sample. If more than one sample was sequenced for a case, the most recent sample was included. Results for recent weeks are incomplete as not all sequencing and bioinformatics analyses were complete at the time of data extraction. Data sources: Public Health Ontario, Hospital for Sick Children, Kingston Health Sciences Centre, Shared Hospital Laboratory, Hamilton Regional Laboratory Medicine Program.

## Methods

### Study design, setting, population, and data sources

Similar to past studies on COVID-19 VE in Ontario[6,23,24], we applied a test-negative design to provincial SARS-CoV-2 laboratory testing, COVID-19 surveillance, COVID-19 vaccination, and health administrative datasets. These datasets were linked using unique encoded identifiers and analyzed at ICES (formerly the Institute of Clinical Evaluative Sciences). The use of the data in this study is authorized under section 45 of Ontario's Personal Health Information Protection Act, and does not require review by a research ethics board.

We included community-dwelling adults aged ≥50 years who had ≥1 reverse-transcription polymerase chain reaction (RT-PCR) test for SARS-CoV-2 between January 2, 2022 and October 1, 2022. We excluded immunocompromised individuals (n = 11,514) and those who received a bivalent mRNA vaccine (n = 2433), Ad26.COV2 (n = 83), or >1 dose of ChAdOx1-S (n = 1043) by the index date (Supplementary Fig. 1). Approximately 95% of individuals received mRNA vaccines (mRNA-1273 or BNT162b2) for all doses. Omicron represented nearly 100% of all positive samples by late January 2022[25–27]. Delta (B.1.617.2) cases identified using whole genome sequencing or based on an S-gene target positive screening result before January 24, 2022 (n = 72) were excluded.

### Outcome and sampling strategy

The case definition was COVID-19-associated hospitalization or death due to, or partially due to, COVID-19. The public health COVID-19 surveillance database data entry guidelines specify that hospitalization data should only be entered for cases who received treatment for COVID-19 while in hospital and/or if their length of stay was extended due to COVID-19[28]. We excluded hospitalizations when specimen collection occurred >3 days after admission and those flagged as being nosocomial. We sampled cases and controls by week of test, thus individuals could enter the study repeatedly, but once an individual became a case, they could not re-enter the study. We employed this sampling strategy to ensure the distribution of time of testing was consistent between cases and controls. Controls had to be symptomatic (Supplementary Text) and test negative for SARS-CoV-2, but may or may not have had a severe outcome. The index date was the earliest of specimen collection, hospitalization, or death.

### COVID-19 vaccination

We classified community-dwelling adults by the number of doses received and time since most recent vaccination relative to the index date. For 2, 3, and 4 doses, we explored up to ≥300 days, ≥240 days, and ≥120 days post-vaccination, respectively. For booster doses of mRNA-1273, a half dose (50 mcg) was recommended for those younger than 70 years and a full dose (100 mcg) for those aged ≥70 years[29].

### Statistical analysis

We used means and proportions to describe our sample by comparing: 1) test-negative symptomatic controls to test-positive Omicron cases who were hospitalized or died; 2) unvaccinated subjects to those who had received 2, 3, or 4 doses; and 3) subjects who had received 2 doses (≥7 days ago) to those who had received 3 or 4 doses. We used standardized differences (SD) to quantify the differences between groups.

Stratified by age group (50–59, 60–69, 70–79, ≥80 years), we used multivariable logistic regression to compare the odds of vaccination in cases to test-negative controls while adjusting for sex, age (continuous), public health unit region, four area-level variables representing different socio-demographic characteristics (household income quintile, essential worker quintile, persons per dwelling quintile, self-identified visible minority quintile), influenza vaccination during 2019–2020 or 2020–2021 (proxy for health behaviors), SARS-CoV-2 infection >90 days prior, number of SARS-CoV-2 tests within

3 months prior to December 14, 2020 (proxy for healthcare workers), comorbidities, receipt of home care services, and week of test (Supplementary Table 1). We estimated the logistic regression models using generalized estimating equations (GEE) with an exchangeable correlation structure since controls could be in a model more than once (13% of controls) leading to non-independence of observations. We calculated both VE and marginal effectiveness using the formula: (1-adjusted odds ratio)*100%.

To examine VE against various Omicron sublineages, we included in our multivariable models an interaction term for time period (BA.1/BA.2-predominant period: January 2, 2022 to July 2, 2022; BA.4/BA.5-predominant period: July 3, 2022 to October 1, 2022) (Fig. 4)[27]. Among these sublineages, the distributions were approximately 50% BA.1 and 50% BA.2 during the BA.1/BA.2-predominant period and 10% BA.4 and 90% BA.5 during the BA.4/BA.5-predominant period[27]. GEE methods were not used in this analysis due to issues with convergence. For the estimates that did converge, GEE and non-GEE estimates and 95% confidence intervals were nearly identical. Additionally, as a sensitivity analysis, we excluded all subjects who had been prescribed Paxlovid within 14 days prior to their index date ($n = 177$) to determine whether treatment impacted VE estimates.

We used SAS version 9.1 (SAS Institute Inc., Cary, NC) for all analyses. All tests were 2-sided and we used $p < 0.05$ as the level of significance. A SDs ≥0.1 were considered clinically relevant.

### Reporting summary

Further information on research design is available in the Nature Portfolio Reporting Summary linked to this article.

## Data availability

The dataset from this study is held securely in coded form at ICES. While legal data sharing agreements between ICES and data providers (e.g., healthcare organizations and government) prohibit ICES from making the dataset publicly available, access may be granted to those who meet prespecified criteria for confidential access, available at https://www.ices.on.ca/DAS (email: das@ices.on.ca).

## Code availability

The full dataset creation plan and underlying analytic code are available from the authors upon request, understanding that the computer programs may rely upon coding templates or macros that are unique to ICES and are therefore either inaccessible or may require modification.

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

## Acknowledgements

This work was supported by funding from the Canadian Immunization Research Network (CIRN) through a grant from the Public Health Agency of Canada and the Canadian Institutes of Health Research (CNF 151944), and also by funding from the Public Health Agency of Canada, through the Vaccine Surveillance Working Party and the COVID-19 Immunity Task Force. This study was supported by Public Health Ontario and by ICES, which is funded by an annual grant from the Ontario Ministry of Health (MOH) and Ministry of Long-Term Care (MLTC). This work was also supported by the Ontario Health Data Platform (OHDP), a Province of Ontario initiative to support Ontario's ongoing response to COVID-19 and its related impacts. J.C.K. is supported by a Clinician-Scientist Award from the University of Toronto Department of Family and Community Medicine. The study sponsors did not participate in the design and conduct of the study; collection, management, analysis and interpretation of the data; preparation, review or approval of the manuscript; or the decision to submit the manuscript for publication. We would like to acknowledge the Canadian Immunization Research Network (CIRN) Provincial Collaborative Network (PCN) Investigators, Public Health Ontario for access to vaccination data from COVaxON, case-level data from the Public Health Case and Contact Management Solution (CCM) and COVID-19 laboratory data, as well as assistance with data interpretation. We also thank the staff of Ontario's public health units who are responsible for COVID-19 case and contact management and data collection within CCM. We thank IQVIA Solutions Canada Inc. for use of their Drug Information File. The authors are grateful to the Ontario residents without whom this research would be impossible. This document used data adapted from the Statistics Canada Postal Code^OM Conversion File, which is based on data licensed from Canada Post Corporation, and/or data adapted from the Ontario Ministry of Health Postal Code Conversion File, which contains data copied under license from

©Canada Post Corporation and Statistics Canada. Parts of this material are based on data and/or information compiled and provided by: MOH, Ontario Health, the Canadian Institute for Health Information, Statistics Canada, and IQVIA Solutions Canada Inc. The analyses, conclusions, opinions and statements expressed herein are solely those of the authors and do not reflect those of the funding or data sources; no endorsement is intended or should be inferred. Adapted from Statistics Canada, Canadian Census 2016. This does not constitute an endorsement by Statistics Canada of this product.

## Author contributions

L.N. obtained the data and conducted the analyses (dataset and variable creation and statistical modeling). R.G. and J.C.K. drafted the manuscript. R.G., L.N., S.A.B., S.E.W., S.N., P.C.A., K.A.B., D.B.F., J.B.G., K.L.S., M.T., K.W., and J.C.K. contributed to the analysis plan, interpreted the results, critically reviewed and edited the manuscript, approved the final version, and agreed to be accountable for all aspects of the work. J.C.K. is the guarantor.

## Competing interests

K.W. is a shareholder and board member of CANImmunize and serves on the data safety board for the Medicago COVID-19 vaccine trial. The other authors declare no other competing interests.
