## [Peer Review File · Nature Communications]

Effectiveness of mRNA COVID-19 vaccine booster doses against Omicron severe outcomesREVIEWER COMMENTS

Reviewer #1 (Remarks to the Author):

The paper discusses the effectiveness of booster doses of monovalent mRNA COVID-19 vaccines against Omicron-associated severe outcomes among adults in Ontario, Canada. The authors used a test-negative design for adults aged ≥ 50 years during the BA.1/BA.2 and BA.4/BA.5 sublineage predominance. The authors showed that booster doses initially restored strong protection against severe outcomes, but protection somewhat declined over time although continue to be high.

The paper is very clearly written and the statistical analysis seems sound and professional. The contribution of the paper is important both strengthening results of previous studies and also generalizing them to the fourth dose and to a long period.

I have some comments.

1. I am somewhat bothered by the differences between the cases and controls. Taking for example the age difference. If most cases are for those who are less vaccinated and are also for the older population (77 vs 66), we should expect higher VE against severe disease just because of the difference in ages. While I understand that age is taken to account by the regression, this is still a major difference.

2. While there is some discussion on the issue of the previous infection, I think that this is a much bigger issue. Assuming that most of the population was infected during the BA.1/BA.2 sublineage predominance, the vaccine effectiveness during the BA.4/BA.5 sublineage predominance should be considered by taking into account the additional protection due to the previous infection. Hence, one should not be surprised that there is a decline in VE as the population is already protected. In the more practical aspect, the question is can you check if someone was previously infected and add that to the analysis?

3. A challenge that was not discussed at all in the paper is the use of antiviral medicines such as Paxlovid. If there is a strong correlation between say those who get the fourth dose and those who get antiviral medicines, the VE is not necessarily related to the vaccine.

4. Related to this comment is the issue of testing which is also mentioned in the paper. As I am not familiar with the testing procedure in Ontario, it will be nice to explain why people get tested by PCR and if this may bias the results. This could be a problem if say, there is a difference related to medical-seeking behavior for both testing and vaccinating.

5. A minor comment regarding how the VE is presented. In your abstract, you write: "Booster doses initially restored strong protection against severe outcomes, but protection declined over time". This makes the impression that the VE is low. However, VE of 75% is still very high. It means that those who are vaccinated are 4 times less at risk compared to the unvaccinated. I would suggest being more careful in writing the conclusion.

Reviewer #2 (Remarks to the Author):

The manuscript, "Effectiveness of mRNA COVID-19 vaccine booster doses against Omicron severe outcomes" submitted by Grewal, et. al. is an important contribution to our understanding of the protection of COVID-19 mRNA vaccines against severe disease during the Omicron period. The stratification of VE results by granular age groups and time since last monovalent COVID-19 mRNA vaccine dose are particularly helpful to inform vaccine policy. This analysis also includes VE estimates on protection against severe disease during the BA.4/5 period for which there remains limited information. The analysis uses a test-negative study design to estimate VE through integration of provincial SARS-CoV-2 laboratory testing, COVID-19 vaccination, and health administrative datasets and is meticulously conducted with thoughtful adjustment for potential confounders. Their results confirm some of the growing evidence during the Omicron period that booster doses of COVID-19 mRNA vaccine offer more protection against severe disease than two

doses. Although booster dose protection appears to wane after vaccination, it remains substantial even 8 months after vaccination among adults aged 50+ years. Additionally, the authors show that VE against severe COVID-19 is lower during BA.4/5 periods than BA.1/2 periods, suggesting increased immune evasion of BA.4/5 and adding to the variable findings on this topic.

Below are a few comments and questions:

1. In the methods, cases are described as having "COVID-19-associated hospitalization or death due to, or partially due to, COVID-19, as specified by data entry guidelines for the public health COVID-19 surveillance database." Could you please elaborate on how the data entry guidelines define a hospitalized case? Elsewhere in the methods, it is explained that community-dwelling adults aged 50+ years who had at least one positive RT-PCR test result for SARS-CoV-2 between January 2 and October 1, 2022 were eligible for inclusion, but the relationship between their SARS-CoV-2 test date and hospital admission date as well as clinical syndrome at admission remain unclear.

2. The authors discuss potential explanations for lower VE during the BA.4/5 period, which include "longer intervals between booster dose receipt and outcomes, increased incidence of undocumented infections, and increased BA.4/5 immune evasion." As stated, the median number of days between booster receipt in the BA.1/2 period and BA.4/5 period was <30 days and therefore unlikely to influence the different VE estimates between variant periods. Regarding undocumented infections, the authors state in the limitations section that rapid antigen tests were the main source of testing after December 31, 2021 (i.e., throughout the period of this analysis), which is reflected in the lower-than-expected proportions of prior infection described in Table 1 (1.1% among cases and 4.9% among controls). As such, the degree to which prior infection is affecting VE results by variant period remains unclear. The additional three months during which BA.4/5 replaced BA.2 may well have resulted in a large number of recent infections that could reduce VE, if the infections occurred disproportionately among unvaccinated or undervaccinated individuals. Was there a surge in COVID-19 cases in Ontario during July-September 2022? Are there any seroprevalence data from Ontario that might inform the potential contribution of prior infection to the VE estimates observed during the BA.4/5 period?

3. In the methods, there is important background related to vaccine dose differences by age. Specifically, the following information is stated, "for booster doses of mRNA-1273, a half dose (50 mcg) was recommended for those younger than 70 years and a full dose (100 mcg) for those aged ≥70 years." Given this differential exposure, it would be helpful to contextualize the VE estimates for booster doses in the 70+ age group by providing the proportion of patients with receipt of mRNA-1273.

4. Minor comment:

Figure S1 indicates that 53,369 test-negative controls were included, whereas lines 45 and 82 indicate that 62,880 test-negative controls were included. Please clarify.

Reviewer #3 (Remarks to the Author):

Dear Authors This is a well written, clear manuscript with interesting and relevant data given the paucity of longer term booster VE in the context of BA4/BA5.

I do have some questions:

The controls used here are symptomatic test negative controls. Its not clear from the MS whether these include hospitalised patients also. With the "decoupling" observed with the omicron wave we noted a divergence in VE estimates where controls were symptomatic negative controls vs admitted (more serious disease)controls. The former therefore tended to overestimate VE. Can you comment on the severity of disease in the control group?

Notwithstanding, the VE estimates are still considerably higher than what we have seen in South Africa and that of data from the US. This may be due to differences in underlying seropositivity rates- is it possible that there are lower seropositivity rates in Canada- perhaps thereby giving rise to a higher VE?

Response to Reviewers: Effectiveness of mRNA COVID-19 vaccine booster doses against Omicron severe outcomes

Reviewer #1:

- 1. I am somewhat bothered by the differences between the cases and controls. Taking for example the age difference. If most cases are for those who are less vaccinated and are also for the older population (77 vs 66), we should expect higher VE against severe disease just because of the difference in ages. While I understand that age is taken to account by the regression, this is still a major difference.**

We agree with the Reviewer that there appear to be substantial differences in some characteristics, including age, between cases and controls overall in Table 1 (original submission). However, all analyses were stratified by age group (50-59, 60-69, 70-79, and ≥ 80 years). To determine whether these differences persisted within each age group, we have revised Table 1 to be stratified by age. Characteristics within age groups are more similar, with an approximate maximum difference in mean age between cases and controls of only 1 year. Additionally, to account for potential confounding effects on VE, all stratified models were adjusted for: sex, age (continuous), public health unit region, four area-level variables representing different socio-demographic characteristics (household income quintile, essential worker quintile, persons per dwelling quintile, self-identified visible minority quintile), influenza vaccination during 2019-2020 or 2020-2021 (proxy for health behaviours), SARS-CoV-2 infection >90 days prior, number of SARS-CoV-2 tests within 3 months prior to December 14, 2020 (proxy for healthcare workers), comorbidities, receipt of home care services, and week of test.

- 2. While there is some discussion on the issue of the previous infection, I think that this is a much bigger issue. Assuming that most of the population was infected during the BA.1/BA.2 sublineage predominance, the vaccine effectiveness during the BA.4/BA.5 sublineage predominance should be considered by taking into account the additional protection due to the previous infection. Hence, one should not be surprised that there is a decline in VE as the population is already protected. In the more practical aspect, the question is can you check if someone was previously infected and add that to the analysis?**

We agree with the Reviewer that previous infection is an important consideration for COVID-19 VE analyses. In our analysis, we excluded participants with a PCR-confirmed infection within the past 90 days and also adjusted for prior infection if the infection was >90 days ago. Additionally, our study sample was restricted to older adults and testing in Ontario was more broadly available for these individuals. Canadian infection-acquired seroprevalence data also suggest that prior infection would have the least impact on VE estimates among adults aged ≥ 60 years compared to other age groups (25–39 years: 75%; 40–59 years: 68%; and ≥ 60 years: 50%)¹. In our Discussion, we note previous infection as a

potential explanation for lower VE against BA.4/BA.5 sublineages and also highlight in our limitations the inability to completely account for previous (undocumented) infection since we did not have access to information on prior infections confirmed only by rapid antigen testing.

Reference:

1. COVID-19 Immunity Task Force. Seroprevalence in Canada. 2022.
<https://www.covid19immunitytaskforce.ca/seroprevalence-in-canada/>

3. A challenge that was not discussed at all in the paper is the use of antiviral medicines such as Paxlovid. If there is a strong correlation between say those who get the fourth dose and those who get antiviral medicines, the VE is not necessarily related to the vaccine.

Paxlovid eligibility in Ontario was initially limited to high-risk individuals (specifically those aged ≥ 70 years and/or unvaccinated or undervaccinated with comorbidities. Initial guidance here: <https://covid19-sciencetable.ca/sciencebrief/clinical-practice-guideline-summary-recommended-drugs-and-biologics-in-adult-patients-with-covid-19-version-10-0/>). Over time, eligibility broadened but remained restricted to those considered at high risk of severe COVID-19 outcomes (i.e., immunocompromised, aged ≥ 70 years, ≥ 60 years with < 3 vaccine doses, and other high-risk adults). Paxlovid receipt could not confound VE estimates since controls (those who test negative for SARS-CoV-2) would have been ineligible for treatment. Moreover, based on the available data (reflecting $\sim 85\%$ of Paxlovid dispensations to outpatients), few cases ($n=177$; 1.6%) in our cohort were prescribed Paxlovid. To assess whether Paxlovid use had any impact on our findings, we completed a sensitivity analysis excluding these individuals. VE estimates were nearly identical to our main analysis findings. We have included this sensitivity analysis to the Supplementary Appendix as Table S8 and incorporated this in the Methods and Results.

4. Related to this comment is the issue of testing which is also mentioned in the paper. As I am not familiar with the testing procedure in Ontario, it will be nice to explain why people get tested by PCR and if this may bias the results. This could be a problem if say, there is a difference related to medical-seeking behavior for both testing and vaccinating.

Although testing is done by both PCR and rapid antigen tests in Ontario, we only have access to results from provincially funded PCR tests. As of December 2021, PCR testing has been restricted to symptomatic individuals at higher risk of COVID-19 severe outcomes, including people who are aged ≥ 70 years, aged ≥ 60 years with < 3 doses of COVID-19 vaccine, aged ≥ 18 years with < 3 doses of COVID-19 vaccine and living with high risk conditions, immunocompromised, at higher risk of severe disease and may be eligible for COVID-19 treatment if tested positive, pregnant, patient-facing healthcare workers/other workers in high-risk settings (e.g., home and community care) and their household members, underhoused, or require a test for clinical management. Additionally, a select few groups are

also eligible for testing whether experiencing symptoms or not. To mitigate biases that may be introduced due to health-seeking behaviours related to testing, we used a test-negative study design, which only includes individuals tested for SARS-CoV-2. We also adjusted for past influenza vaccination as a proxy for health behaviours in all our models.

- 5. A minor comment regarding how the VE is presented. In your abstract, you write: “Booster doses initially restored strong protection against severe outcomes, but protection declined over time”. This makes the impression that the VE is low. However, VE of 75% is still very high. It means that those who are vaccinated are 4 times less at risk compared to the unvaccinated. I would suggest being more careful in writing the conclusion.**

We understand and appreciate the Reviewer’s perspective on our manuscript’s conclusions. Although we agree that VE of 75% would be high against infection, we believe it may be suboptimal against severe outcomes, particularly among older, more vulnerable adults. Additionally, when stratified by sublineage period, VE was substantially lower during the more recent BA.4/BA.5-predominant period. For example, though the overall VE was 75% 180-239 days after a third dose among individuals aged ≥ 80 years, the VE during the BA.4/BA.5-predominant period across the same time period since vaccination was only 52%. We have clarified this interpretation in the Conclusion of the Abstract.

Reviewer #2:

- 1. In the methods, cases are described as having “COVID-19-associated hospitalization or death due to, or partially due to, COVID-19, as specified by data entry guidelines for the public health COVID-19 surveillance database.” Could you please elaborate on how the data entry guidelines define a hospitalized case? Elsewhere in the methods, it is explained that community-dwelling adults aged 50+ years who had at least one positive RT-PCR test result for SARS-CoV-2 between January 2 and October 1, 2022 were eligible for inclusion, but the relationship between their SARS-CoV-2 test date and hospital admission date as well as clinical syndrome at admission remain unclear.**

The COVID-19 surveillance data entry guidelines for provincial surveillance specify that hospitalization data should only be entered for cases who received treatment for COVID-19 while in hospital and/or if their length of stay was extended due to COVID-19. We have included this definition in the Methods section of the manuscript. The index date for inclusion into the study was the *earliest* of specimen collection for testing, hospitalization, or death. If an individual was tested >3 days after they were admitted to the hospital, they were excluded from the analysis. All clinical syndromes requiring COVID-19-associated hospitalization were included, however, those flagged as nosocomial were excluded.

- 2. The authors discuss potential explanations for lower VE during the BA.4/5 period, which include “longer intervals between booster dose receipt and outcomes, increased incidence of undocumented infections, and increased BA.4/5 immune evasion.” As stated, the median number of days between booster receipt in the BA.1/2 period and BA.4/5 period was <30 days and therefore unlikely to influence the different VE estimates between variant periods. Regarding undocumented infections, the authors state in the limitations section that rapid antigen tests were the main source of testing after December 31, 2021 (i.e., throughout the period of this analysis), which is reflected in the lower-than-expected proportions of prior infection described in Table 1 (1.1% among cases and 4.9% among controls). As such, the degree to which prior infection is affecting VE results by variant period remains unclear. The additional three months during which BA.4/5 replaced BA.2 may well have resulted in a large number of recent infections that could reduce VE, if the infections occurred disproportionately among unvaccinated or undervaccinated individuals. Was there a surge in COVID-19 cases in Ontario during July-September 2022? Are there any seroprevalence data from Ontario that might inform the potential contribution of prior infection to the VE estimates observed during the BA.4/5 period?**

We agree that undocumented infections may have influenced VE estimates over time, given the rise in infections across the study period. However, in our Discussion, we noted that VE declined considerably faster as *time since vaccination* increased during the relatively brief BA.4/BA.5-predominant period (only 3 months) compared to the BA.1/BA.2-predominant period, suggesting that bias from undocumented prior infections is unlikely to account *entirely* for the differences. Additionally, although infection-acquired seroprevalence in Ontario increased from 50% to 63% between July and the beginning of October 2022 (BA.4/BA.5-predominant period), infection-acquired seroprevalence saw a much larger increase from January 2022 to the beginning of July 2022 (BA.1/BA.2-predominant period) with an increase from 6% to 50% (see Figure below [infection-acquired seroprevalence estimates in blue]).¹ It is also important to note that the figure provided is for all ages (and who are blood donors), and in Canada, infection-acquired seroprevalence is even lower among adults aged ≥ 60 years (the majority of our study sample) at 50% by October 2022.¹ We have included more information on infection-acquired seroprevalence in Ontario and acknowledged the limitation of undocumented infection in the Discussion.

Reference:

- COVID-19 Immunity Task Force. Seroprevalence in Canada. 2022. <https://www.covid19immunitytaskforce.ca/seroprevalence-in-canada/>

- In the methods, there is important background related to vaccine dose differences by age. Specifically, the following information is stated, “for booster doses of mRNA-1273, a half dose (50 mcg) was recommended for those younger than 70 years and a full dose (100 mcg) for those aged ≥ 70 years.” Given this differential exposure, it would be helpful to contextualize the VE estimates for booster doses in the 70+ age group by providing the proportion of patients with receipt of mRNA-1273.

Among those aged 70-79 years, 36% and 31% of individuals who received dose 3 and 4, respectively, received mRNA-1273. Among adults aged ≥ 80 years, these proportions were 34% and 37%, respectively. We have included these findings in the Results section.

- Figure S1 indicates that 53,369 test-negative controls were included, whereas lines 45 and 82 indicate that 62,880 test-negative controls were included. Please clarify.

In Figure S1, 53,369 refers to the unique number of test-negative *individuals* included in the analysis whereas 62,880 in the Results refers to the number of *tests* included for controls. Controls could re-enter models, meaning they could contribute multiple tests, until (and if) they became a case. We have clarified this in the Abstract and Results.

Reviewer # 3:

- The controls used here are symptomatic test negative controls. It’s not clear from the MS whether these include hospitalised patients also. With the "decoupling" observed with the omicron wave we noted a divergence in VE estimates where controls were symptomatic negative controls vs admitted (more serious disease) controls. The former

therefore tended to overestimate VE. Can you comment on the severity of disease in the control group?

Due to lags in the availability of health administrative data (required for ascertaining hospitalization status of test-negative controls but not for test-positive cases, which are recorded in the CCM database), we were only able to determine whether controls were hospitalized or not between the period of January 2, 2022 and July 31, 2022. During this period, 22% of symptomatic controls included in our analysis were hospitalized. Based on data availability, we feel using test-negative controls who are not necessarily hospitalized was the most appropriate reference group for ascertaining VE against severe outcomes.

2. Notwithstanding, the VE estimates are still considerably higher than what we have seen in South Africa and that of data from the US. This may be due to differences in underlying seropositivity rates - is it possible that there are lower seropositivity rates in Canada - perhaps thereby giving rise to a higher VE?

COVID-19 seropositivity estimates in Ontario increased considerably from December 2021 to October 2022, with infection-acquired seroprevalence estimated at approximately 63% among all ages by early October 2022.¹ However, across Canada, infection-acquired seroprevalence has varied considerably by age, with the lowest estimates among adults aged ≥ 60 years (50% by October 2022), the majority of our study sample. These estimates are lower than in the US where infection-acquired seroprevalence for the general population ranged between 73% and 97% across different contiguous US jurisdictions by mid-August 2022.² Similarly, a study in South Africa found that infection-acquired antibodies had already reached 70% and 60% in an urban and rural community, respectively, by November 2021.³ Another potential explanation for differences in VE estimates is that our analyses were stratified by age group and restricted to adults aged ≥ 50 years whereas analyses in South Africa and the US were among all adults aged ≥ 18 years. Comparisons with other jurisdictions are also challenging due to differences around other study elements, such as study design, population characteristics, outcomes (i.e., different severe outcomes and definitions) and exposures (i.e., different time periods since vaccination), vaccines (e.g., South Africa study only assessed VE for the Pfizer vaccine), and observation periods (e.g., South Africa's BA.4/BA.5 period began much earlier than in North America).

References:

1. COVID-19 Immunity Task Force. Seroprevalence in Canada. 2022. <https://www.covid19immunitytaskforce.ca/seroprevalence-in-canada/>
2. National Institutes of Health. COVID-19 SeroHub. 2022. <https://covid19serohub.nih.gov/>
3. Kleynhans, et al. SARS-CoV-2 seroprevalence after third wave of infections, South Africa. *Emerg Infect Dis.* 2022;28(5):1055-1058.

REVIEWERS' COMMENTS

Reviewer #1 (Remarks to the Author):

I am satisfied with the revision. While this is an observational study and has some limitations, the authors succeeded in addressing many obstacles through careful analysis, and this is a very thorough study.

Reviewer #2 (Remarks to the Author):

Thank you for these responses. I have no further questions.

Reviewer #3 (Remarks to the Author):

Dear Editors

I have reviewed the rebuttals and updated MS. The study has a number of limitations but these have been adequately explained by the authors and i am satisfied that the paper is publishable and the study's limitations are now adequately represented.